# A Portable Device for I–V and Arrhenius Plots to Characterize Chemoresistive Gas Sensors: Test on SnO_2_-Based Sensors

**DOI:** 10.3390/nano13182549

**Published:** 2023-09-12

**Authors:** Michele Astolfi, Giulia Zonta, Sandro Gherardi, Cesare Malagù, Donato Vincenzi, Giorgio Rispoli

**Affiliations:** 1Department of Physics and Earth Science, University of Ferrara, 44122 Ferrara, Italy; michele.astolfi@unife.it (M.A.); giulia.zonta@unife.it (G.Z.); malagu@fe.infn.it (C.M.); donato.vincenzi@unife.it (D.V.); 2SCENT S.r.l., 44124 Ferrara, Italy; gherardi@fe.infn.it; 3Department of Neurosciences and Rehabilitation, University of Ferrara, 44121 Ferrara, Italy

**Keywords:** gas sensor, chemoresistivity, sensor device, current-voltage characteristics, Arrhenius plot, metal-oxide, portable device, sensor calibration

## Abstract

Chemoresistive nanostructured gas sensors are employed in many diverse applications in the medical, industrial, environmental, etc. fields; therefore, it is crucial to have a device that is able to quickly calibrate and characterize them. To this aim, a portable, user-friendly device designed to easily calibrate a sensor in laboratory and/or on field is introduced here. The device comprises a small hermetically sealed chamber (containing the sensor socket and a temperature/humidity sensor), a pneumatic system, and a custom electronics controlled by a Raspberry Pi 4 developing board, running a custom software (Version 1.0) whose user interface is accessed via a multitouch-screen. This device automatically characterizes the sensor heater in order to precisely set the desired working temperature, it acquires and plots the sensor current-to-voltage and Arrhenius relationships on the touch screen, and it can record the sensor responses to different gases and environments. These tests were performed in dry air on two representative sensors based on widely used SnO_2_ material. The device demonstrated the independence of the Arrhenius plot from the film applied voltage and the linearity of the I–Vs, which resulted from the voltage step length (1–30 min) and temperature (200–550 °C).

## 1. Introduction

Metal-oxide (MOX) gas sensors, thanks to their high sensitivity (attaining a detection limit up to tens of ppb) [1,2,3,4], low power consumption, fast response (within 10–20 min), and recovering time (typically 15–30 min) [4,5], are widely used in different application fields, such as environmental [6,7,8], industrial [9], agri-food [10,11,12,13,14], medical [15,16,17,18,19,20,21], etc. MOX sensor sensing principle is based on chemoresistivity (i.e., their electrical resistance changes as a function of the chemical reactions between the surface and a gaseous analyte), which is enhanced by nanostructured manufacturing the sensing film (i.e., the film is made up of an interconnected network of semiconductor nanograins). The latter can be obtained through several methods [22], such as sputtering [23,24], chemical vapor deposition [22,25], screen-printing deposition of a nanostructured paste [15,26] (synthesized for instance by means of the sol-gel technique [27,28]), etc. Despite their high sensitivity, these sensors exhibit a low selectivity to specific compounds if used singularly [4]; this limit can be overcome to some extent by combining them into arrays of two or more sensors and carefully calibrating each one, so as to optimize the array for specific applications (e.g., detection of a particular gas target or mixture). The large number of possible applications for these sensors makes it necessary to have a system able to easily calibrate and test them, in order to select the most suitable ones for a specific purpose. Usually, most laboratories perform on-site calibrations only by using large, bulky, and costly set-ups, which are often not suitable for performing on-field calibrations (of environmental sensors, for instance), and require trained personnel. Nowadays, much research is focused on the development of user-friendly and portable devices for gas analysis and the detection of their impurities by employing different technologies [29,30,31,32]. In this work, a novel, portable, and user-friendly device is introduced (so far in the form of a hand-made demonstrator), which is entirely designed, produced, and assembled at the Sensor Laboratory (SL) of the Department of Physics and Earth Sciences of the University of Ferrara and is able to characterize and/or calibrate a single MOX sensor in a laboratory and/or field, and able to analyze single or mixed gases. The electronics and pneumatics employed for this device rely on the SL team’s experience acquired in manufacturing devices for medical screening and environmental monitoring purposes [17,18,19,21,33]. The device was also optimized to be remarkably versatile and low power consuming without any loss of reliability, and is therefore capable of handling most of the currently available MOX sensors. The device was conceived for the following applications: (i) automatic calculation and tuning of the heater resistance to set precisely the desired working temperature (WT); (ii) rapid construction of the sensor current-to-voltage (I–V) and Arrhenius curves [5,34,35]; (iii) acquisition of the sensor signal to different gases and environments and the sensor response calculation. To validate the device performances, all these tests were performed in dry air on both the SnO_2_ and SnO_2_ + 1%Au representative sensors, because of the wide use of the tin-oxide semiconductor material in the sensor field.

## 2. Materials and Methods

### 2.1. Chemoresistive Sensors Fabrication

The representative sensors employed here are based on tin-oxide semiconductor nanostructures decorated or not with gold nanoparticles (SnO_2_ and SnO_2_ + Au 1%), because of the high versatility and popularity of SnO_2_-based materials [29,36]. These sensors were entirely designed, produced, and assembled at SL, through commonly employed techniques for thick-film MOX sensors production. The molar addition of 1% Au in the SnO_2_ material leads to an improvement of material stability, sensitivity, and selectivity, and increases its conductance with respect to pure SnO_2_ [3]: noble materials work as catalysts, thereby improving the molecular oxygen dissociation at the sensor surface during the sensing process [36]). Sensors are typically made of the following three components (Figure 1):the substrate: an alumina-made insulating layer hosting two interdigitated comb-shaped gold electrodes, connecting the sensor to the readout circuit;the sensing material (or active material): a porous thick film (thickness ~20 µm) of interconnected MOX nanoparticles;the heater: a platinum meander aimed to heat the sensor to the proper WT by controlling the current flowing through it.

The MOX nanopowder, comprising the sensing material, was synthesized through sol-gel technique, in a similar way as indicated in [28], then converted into viscous paste through the addition of organic vehicles and a glass frit (a mixture of glassy silicon oxides, which are crucial to optimize the adhesion between the film and the substrate). The paste was then printed onto the alumina substrate by means of a screen-printing machine (Aurel C920, Modigliana, Italy) and thermally treated through drying (at about 100 °C) and firing (at temperatures up to 850 °C). The former process eliminates the residual volatile additives, the latter one uniforms and stabilizes the nanograin’s dimensions and permanently fixes the film to the substrate. Finally, the sensor was welded by thermo-compression (bonding technique) on a four pin TO-39 socket, connecting it to the readout circuit [5].

The SnO_2_ nanoparticles were synthesized by dissolving Sn(II) ethylhexanoate in a hydro-alcoholic solution (H_2_O/2-propanol mix) and stirring at room temperature. Then, an aqueous solution of AuBr_3_ was added to this mixture to obtain a 100:1 SnO_2_ to Au molar ratio. The precursor was hydrolyzed by adding 0.15 M of HNO_3_ and then calcinated in air for 2 h at 650 °C to obtain the SnO_2_ (Au 1%) nanostructured powder composed of grains that ranged between 50–200 nm in size. All materials were purchased from Sigma–Aldrich and were used without further purification, as reported in [37]. Finally, the semiconductor paste was printed onto the front side of an alumina substrate and equipped with comb-shaped gold-electrodes and a heather (platinum meander; Figure 1b) printed on the backside to heat up the sensor to the desired WT (generally ranging from 300 to 600 °C) [5].

### 2.2. Single Sensor Device and Set-Up

The portable device for I–V and Arrhenius plots that characterizes chemoresistive gas sensors (20 × 25 × 15 cm^3^ weighting about 2.5 kg) hosts an electronic system powered by a 24 W power supply, including a sensor board controlled by a Raspberry Pi 4 and a cylindrical sensor chamber. The Raspberry Pi 4 works as the device-computing unit, providing the serial communication (I^2^C) to the electronics components (i.e., digital-analog converter, DACs; analog-digital converter, ADCs; etc.; Figure 2) and managing the entire device through the external touch screen by means of a dedicated software (written in Python language; Version 1.0). The sensor board comprises one unit responsible for the sensor heating and a second unit dedicated to the sensor signal acquisition.

The sensor heating is attained by means of a 12-bit DAC (DAC1, MCP4725, Microchip Technology Inc., Chandler, AZ, USA) that supplies a reference voltage (up to 10 V) to a LDO (low drop-out voltage regulator, LT3080, Analog Devices, Wilmington, MA, USA) necessary to boost up the DAC1 current feeding the heater (Figure 2). The voltage drop across the resistor *R_c_* (which is read by one of the four 16-bit ADC channels; ADS1115, Texas Instruments, Dallas, TX, USA) in series with the heater, is used to monitor the actual current flowing through the latter, to fine-tune the DAC1 output to reach the desired WT (see Section 2.3). This electronic device is very precise and at the same time over dimensioned, since it can heat the sensor to temperatures higher than one thousand degrees Celsius (though it is strongly recommended to not heat this sensor type above its firing temperature, typically between 600–900 °C).

The signal acquisition circuit (Figure 3) applies a variable voltage $V_{i}$ (0–10 V) to the sensor film by means of a second 12-bit DAC (DAC2, MCP4725; Figure 2), while an inverting operational amplifier (IOA) returns a voltage SIGNAL (Equation (1)) as a function of time inversely proportional to the sensor film resistance $R_{s}$:(1)$$SIGNAL=-\frac{R_{f}}{R_{s}}V_{i}$$
where $R_{f}$ is the IOA feedback resistance. The managing electronics can easily handle sensors with a film resistance exceeding 3 GΩ (therefore covering most of the MOX sensors currently available) and also generating very low signal amplitudes, thanks to its very low noise and reliability. The SIGNAL was sampled and digitized by a second 16-bit ADC channel of the ADS1115 integrated circuit, and acquired and plotted in real time by means of the custom software. In order to acquire the I–V relationships, it is necessary to measure the current $i_{f}$ flowing through the film: since the film voltage $V_{i}$ is related to the film current intensity by
(2)$$V_{i}=R_{s}i_{f}$$
the absolute value of $i_{f}$ can be calculated by combining Equations (1) and (2):$$\left|i_{f}\right|=\frac{SIGNAL}{R_{f}}$$

The sensor chamber is a hermetically sealed aluminum cylinder having both external height and diameter of 7 cm, 1 cm wall thickness, and an openable head. The chamber hosts a gas sensor socket, a humidity/temperature sensor (SHTC3, Sensirion AG, Stäfa Switzerland), an 8-pin electronic connector to couple the tested and the humidity/temperature sensors to the external electronics, and two swift pneumatic connectors for 4 mm-Teflon tubes, working as gas inlet and outlet (Figure 4).

The device is completed with a pneumatic system that conveys the gas to the sensor chamber from, for instance, an external cylinder, and regulates the gas flux though a mass flow controller (GF40 series, Brooks Instruments, Hatfield, PA, USA) and a multitouch external screen (Figure 5). Other gas lines, when equipped with a mass flow controller, could be merged by means of a multiple connector to convey different interfering gases to the inlet of the sensor chamber (Figure 5); the humidity of a particular gas line could be arbitrarily changed by using a bubbler [3,5].

Furthermore, it is possible to convey the environmental air to the sensor chamber through a supplementary gas line comprising an optional internal pump, a carbon filter (to reduce the air relative humidity variation), and a 0.2 μm filter to remove most of the particulate contaminating the incoming air.

### 2.3. WT Determination

For the correct sensor functioning, it is important to heat its film to a specific WT [5], depending on the material type, the environmental atmosphere, the target gas, or gas mixture to be analyzed. The optimal WT at which the sensor exhibits the best performances (sensitivity, repeatability, detection limit, recovery time, etc.) for a certain application can be experimentally determined after calibration tests and usually ranges from 300 to 650 °C for most of the MOX sensors. To precisely heat the sensor to a proper WT, it is necessary to apply an appropriate voltage to the platinum heater, generating a current (measured by the voltage drop across *R_c_*; Figure 2) that heats the sensor through the Joule effect. The heater resistance, calculated by dividing the applied voltage and the measured current, increases with temperature, according to
(3)$$R_{T}=R_{0}\left(1+\alpha T+\beta T^{2}\right)$$
where $R_{T}$ is the heater resistance at the generic temperature T (in °C), $R_{0}$ is the resistance at 0 °C, while *α* and *β* are constants particular to the heater material ($\alpha =0.003263{\;}^{{}^{\circ}}C^{-1}$ and $\beta =-6.6668\times 10^{-7}\;{}^{\circ}C^{-2}$). $R_{0}$ is the unique parameter of Equation (3) that must be known to calculate $R_{T}$ (by the software; Version 1.0), being calculated through Equation (4) (obtained by rearranging Equation (3)):(4)$$R_{0}=\frac{R_{h}}{1+\alpha T_{h}+\beta T_{h}{}^{2}}$$
where $R_{h}$ is the heater resistance measured at the reference temperature $T_{h}$. Therefore, it is possible to precisely control the sensor temperature by applying an appropriate voltage $V_{T}$ that generates a current intensity I flowing through the heater given by
(5)$$I=\frac{V_{T}}{R_{T}}$$

All the above procedures (i.e.,$T_{h}$ and $R_{h}$ measurement, $R_{0}$ calculation, and $V_{T}$ setting) could be executed manually or automatically by the custom program (see Section 2.5).

The chamber temperature and humidity measured by the dedicated sensor were kept at almost constant values (34–36 ± 0.2 °C and 2 ± 1%, respectively) during the measurements, while the dry air or the test gas flow was kept constant by means of a mass flow controller (Figure 5).

### 2.4. I–V Characteristic Curves Determination

The study of the I–V curves of a material is crucial to understand its electrical behavior and hence to monitor its expected electrical functioning. For instance, it can verify whether I–Vs exhibit a hysteresis or if they have a particular shape (e.g., linear, exponential, power law, etc.). These experiments could also enlighten possible film structural changes (long lasting or not) induced by the electrical field variations. The voltage region in which there is no hysteresis allows easy control of the IOA gain (Equation (1)), aside from *R_f_* (which is often inconvenient), also by changing the film applied voltage, which can be easily adjusted. These tests are generally performed in a laboratory, following the sensor manufacturing and assembly, and/or before the sensor replacement in an operative device. The portable device described here obtains I–V curves with high accuracy, rapidity, and, if necessary, directly on field as follows. After the sensor WT determination (see Section 2.3) and the sensor stabilization at this temperature (i.e., when the sensor SIGNAL is constant in a reference atmosphere), the film voltage can be changed, manually or automatically, from 0 to 10 V in arbitrary steps of an arbitrary duration, while the device records the current intensity at the end of each step. In automatic mode, the device performs the I–V curve with a number of voltage steps and duration previously entered by the user through the software graphical interface (Version 1.0; Figure 5); the voltage steps can be applied in increasing and/or decreasing directions. At the end of the I–V curve acquisition, the data are stored in a .txt file, which can be used by any analysis or plotting software (Version 1.0).

### 2.5. The Managing Software

The entire device is controlled by means of a custom software (written in Python language; Version 1.0) running on a Raspberry Pi 4 developing board computer. A multitouch-screen allows the user to interface with the software, which handles, monitors, and plots all the sensor parameters in real time. The user can interact with the device by clicking on the desired button and entering values and notes by typing on an on-screen virtual keyboard. All the commands and plots are organized in three main pages (Figure 6 and Figure 7) named Heaterpage, IVpage, and Signalpage. The first one (Figure 6) manages the sensor temperature settings, allowing adjustment to the desired WT manually or automatically, collecting the data for the Arrhenius plots by clicking on the corresponding button, and displaying the chamber internal temperature (“T”) and humidity (“H”). First of all, on this page (Figure 6, top panel), the “Create Sensor” button allows the user to enter the sensor name, “Name:”, the senor “Code” (univocally identifying it), and some arbitrary “Notes” that may keep track of the experiment type and conditions. This page displays in real time the “Voltage” applied to the heater, the measured “Current” flowing through it, and the corresponding heater resistance “Rh”. In order to characterize the sensor heater, the device measures $R_{h}$ at room temperature $T_{h}$ by applying a very small voltage to the heater and measuring the corresponding current. By exploiting $R_{h}$ and $T_{h}$, the software calculates “R0” (according to Equation (3)), that is required to calculate the heater resistance corresponding to the desired WT. Once “R0” is computed, the software calculates the theoretical heater resistances (R_T_s) related to a list of arbitrary preselected temperatures (from 200 to 650 °C in 50 °C step here) exploiting Equation (2), and generates the temperature table (Figure 6, upper panel) where the temperatures and the “Rh”s are inserted in the first and second row (black and red numbers), respectively. At this point, by trial and error, the user can find the appropriate value to enter in “H.DAC” field to obtain the desired WT, by matching “Rh” and the value of the second row of the table (red numbers) related to the desired WT (first row of the table). Moreover, it is possible to set the WT directly by clicking on the corresponding temperature label (that turns red): this starts the software algorithm that automatically sets the correct “H.DAC” value (within an arbitrary error set via software, being currently 2%) by successive approximations.

The “Arrhenius plot” button opens a subpage (Figure 6, bottom panel), where the starting (“From”) and the final (“to”) DAC1 values (between 0 and 4096) can be entered, which set the corresponding WTs; the number of temperature steps in this range can be entered in the “N.Steps” field and the duration of each step is entered in “St.Time”; these four entries cannot be modified during the Arrhenius data acquisition to avoid possible user errors. The Arrhenius data acquisition is started by clicking the “Start” button and forced to stop with the “Cancel” one. Finally, the user can enter/update the value of “Rfeed” after the physical insertion or replacement of $R_{f}$ in a dedicated connector (this could be done during the measurements). At the end of each temperature step, ”Rh”, “Vs” (which is the sensor SIGNAL), and “Rf” data are reported in the appropriate list box field, ready to be stored in a .txt file by clicking on “SvD” (Save Data) button. “Rm” (Remove) and “RmA” (Remove All) buttons delete a selected or all the data items in the list box, respectively. The program also plots SIGNAL vs. time, to ensure the operator that the data are acquired correctly and to prevent a possible signal saturation.

The “IVpage” (Figure 7 upper panel) is composed of three main frames: the first one is for sensor film management, a second one is to preview the sensor SIGNAL vs. time (the same plot is shown in a larger frame in the “Signalpage”, Figure 7 lower panel, where the plot can be stopped and erased by the “STOP” and “RESET” buttons), and a third one includes a list box stacking the acquired data. The list box columns include the film voltage “V (V)”, the film current “I (microA)”, the feedback resistor “Rf” (entered in Ω), and the sensor chamber temperature “T (°C)” and humidity “H (%)”. The data stored in the list box can be saved in a .txt file by clicking the “Save Data” button. “Remove” and “Remove All” buttons have the same function of “Rm” and “RmA”. The sensor film management frame includes an entry (“DAC”) to set the DAC2 value corresponding to the film voltage, a further “Rfeed” entry (in Ω, that is coupled to the one in the “Arrhenius plot” page), the real time-updated film voltage (“V”), current (“I”), and resistance (“R”). This frame also includes the “ACQUIRE I–V” and “AUTO ACQUISITION” buttons to acquire the data manually (each button press acquires a single I–V curve data) or automatically (through a dedicated software algorithm), respectively.

In brief, the managing software (comprising more than 1600 program codes, Figure 8) is organized with a “Main Thread”, flanked by a further thread (“Thread 1”) that is aimed to continuously read the data from the electronics and to update the software variable values (it is the sole thread that has access to the electronic board in order to avoid any race condition). A second thread (“Thread 2”) is used to plot and save the sensor voltage (acquired from “Thread 1”) in real time to execute the Arrhenius plot (Arrhenius Plot Page) or to simply execute the voltage plotting vs. time (“SignalPage”). Finally, a third thread (“Thread 3”) performs the voltage changes to the sensor film, plots the I–V points, and stores the I–V data in a .txt file.

## 3. Results and Discussion

An important sensor characterization is provided by the Arrhenius plot that describes how the sensor conductance (G) changes with the sensor heating temperature (T) in the presence of a defined target gas (Figure 9). In this plot, it is possible to distinguish three regions: the first one between 2.2 and 1.8 (1000/°K), the second one between 1.1 and 1.4, and the last one between 1.4 and 1.0.

In the first region, it is reasonable to assume that the almost linear conductance increases in the logarithmic scale with T (i.e., G varies exponentially on 1/T with negative exponent), depending on the progressive increase of the number of conduction band electrons excited by thermionic effect. Indeed, the atomic oxygen (O) is more reactive in respect to the molecular one (O_2_) and prevails at those low temperatures O_2_, leading to a small number of electrons seized from the film conduction band. In the second region, the conductance is roughly constant despite the temperature increase, since the further thermionic growth of the number of conduction band electrons is compensated by an almost equal number of electrons captured by the atomic oxygen (whose number is increased in respect to the first region because of the increased temperature) ionosorbed on the sensor surface. In the third region, the oxygen (mainly in its atomic form), seizes a number of conduction band electrons at a rate smaller than the one produced by thermal excitation. The Arrhenius plot was negligibly affected by the change of the applied film voltage (from 2.7 to 8.4 V, Figure 9, upper panel). To test the reproducibility of these measurements, the Arrhenius plot at each voltage was repeated three times, and the nine resulting curves were averaged and plotted with their standard errors (Figure 9, lower panel). In conclusion, the electrical field applied to the sensor film nanograins does not influence the trend with which the rate of the electrons drift along the film changes with the temperature [34,38], i.e., the applied voltage does not affect the film conductance at an arbitrary WT.

In order to investigate the influence of the Au addition to the sensor paste, the Arrhenius plots at the various film voltages were also performed on the pure SnO_2_ material (Figure 10, red lines).

The Au addition increased the film conductance by about an order of magnitude in the range of 320–400 °C, and slightly widened the plateau phase (i.e., where the conductance is almost constant with the temperature, making the sensor more stable in a wider range around the WT of this material). To further characterize the electrical performance of a chemoresistive sensor film, it is crucial to study the I–V behavior of its material. On this basis, the current flowing through the sensor film (SnO_2_ + 1%Au) was measured for voltages ranging from 0 to 10.4 V in ten steps of 1.04 V (Figure 11) at three different key temperatures (200, 350, and 550 °C) in ascending and descending directions. These temperatures were carefully selected in order to measure the IVs at the center of the three Arrhenius plot representative regions (Figure 9). This particular sensor film exhibited a negligible hysteresis and a slight non-linearity for voltages above 9 V, although it was challenged with voltages exceeding the ones usually employed. In any case, this slight non-linearity is not present in the averaged IVs (Figure 12), making this sensor kind particularly suitable for applications where the film I–V linearity is strongly required. Since the typical MOX sensors WT ranges between 300 and 600 °C, the I–Vs performed at 200 °C are reported only for completeness and not for practical reasons.

In order to estimate how long it took for the sensor film to attain a steady state current once the voltage step is applied (i.e., whether there were time-dependent changes in film resistance), the latter lasted 1, 10, and 30 min, and the corresponding I–Vs were plotted by measuring the current intensity just at the end of each step (Figure 11). The relative high voltage steps and their duration aimed to verify whether the strong and long lasting electric fields applied to the sensor film could produce structural changes that could modify the sensor performances (here in dry air). In the case of SnO_2_ + 1%Au, the ascending and descending I–V curves were almost linear and superimposed at all the tested temperatures and the step durations considered. Moreover, the almost overlapping I–Vs at 1, 10, and 30 min indicated that the applied voltage did not cause long-lasting changes in the film structure. In order to check the reproducibility of the above measurements, they were repeated three times for each step duration (1, 10 and 30 min) at the three key WTs (200, 350, and 550 °C). For clarity, in Figure 12 (top panel) the average I–Vs and the related standard errors are shown at 350 °C only, which is the usual WT for this sensor type. The resulting curve shows a good linearity and a negligible hysteresis. Furthermore, the standard error was always <4%, and once grouping together all the data point at the same voltage, the standard error fell down to <3% and the I–Vs were well fitted by a straight line (Figure 12, bottom panel).

The influence on the I–Vs waveform of the Au addition to the sensor paste was evaluated by repeating all the I–V protocols of Figure 12 on the pure SnO_2_ material (Figure 13).

The average I–Vs and the related standard errors for SnO_2_ material at 350 °C only, which is the usual WT for this sensor type, shows again a good linearity and a negligible hysteresis; the standard error was always <4% and it fell down to <3% once grouping together all the data points at the same voltage (Figure 13, bottom panel). The resulting conductance at 350 °C resulted about one order of magnitude smaller than the SnO_2_ + 1%Au sensor heated at the same temperature, which is consistent with the Arrhenius plot outcomes (Figure 10).

Similar results were obtained at 200 and 550 °C for both SnO_2_ + 1%Au and pure SnO_2_ materials; all the above measurements could be repeated in the presence of various gases of interest or a mixture of them by using the set-up described in Section 2.2.

## 4. Conclusions

The user friendly device introduced here, developed to handle most of the currently available MOX sensors, was able to rapidly test and calibrate two representative SnO_2_-type sensors in dry air. The SnO_2_ and SnO_2_ + 1%Au sensors, based on the widely used tin-oxide material, were chosen to test and validate the device. The latter precisely calculated and set the desired sensor WT and it acquired and plotted the sensor Arrhenius plots and I–V relationships on a touch screen. The Arrhenius plots were performed automatically for each one of the two sensors at different film voltages (ranging from 2.7 to 8.4 V) in dry air, demonstrating the independency between them and the film-applied voltage, resulting as well superimposable. This indicates that the film voltage does not affect the chemical reactions occurring on the sensor surface. The I–Vs in dry air of the two sensors were performed at three key WTs (200, 350, and 550 °C) that were identified in the center of the three Arrhenius plots representative regions. In particular, the I–Vs of these sensors were almost linear with a negligible hysteresis effect at all three selected temperatures and for long-lasting voltage applications. Based on the I–Vs and the Arrhenius plots, the Au addition significantly increased the SnO_2_ conductance and enlarged the Arrhenius plot plateau region without any alteration of the I–Vs linearity. The lack of time-dependent changes of sensor film resistance following a voltage change also allows control over the sensor signal output amplification by adjusting the film applied voltage. Finally, this study showcases the suitability of SnO_2_-based sensors when employed in applications where the film conductance linearity is required at different WTs, independently from the duration of the applied voltage.

## Figures and Tables

**Figure 1 nanomaterials-13-02549-f001:**
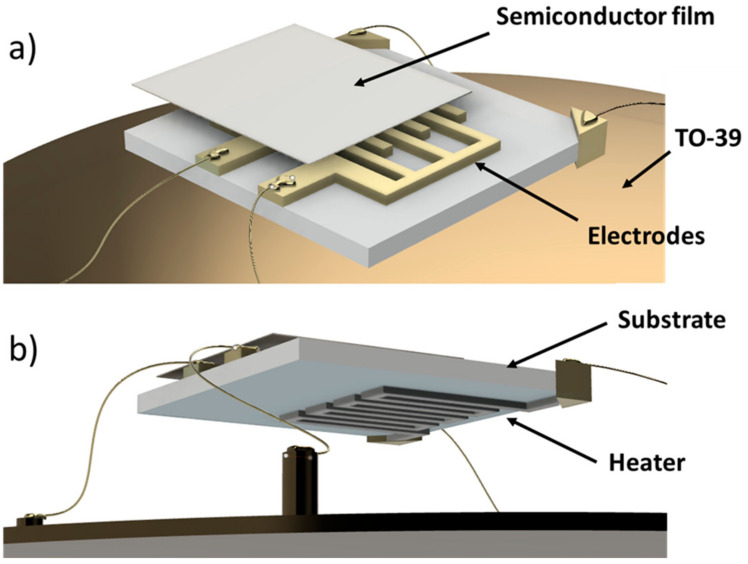
3D sketches of a sensor. (**a**) Sensor upper view: the semiconductor film is printed onto the substrate (connecting the two comb-shaped gold electrodes); the sensor is bonded with golden wires to a TO-39 socket; (**b**) sensor bottom view: the heater (platinum meander) is shown on the substrate backside.

**Figure 2 nanomaterials-13-02549-f002:**
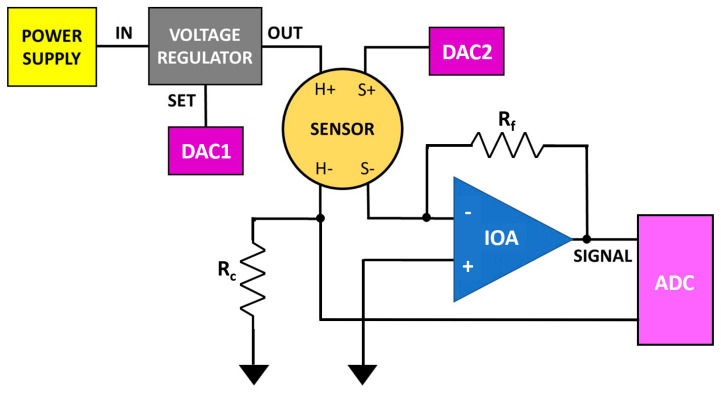
Block-scheme of the device electronics. H+ and H−, S+, and S− are the heater (driven by DAC1) and the sensor film pins, respectively; the voltage drop across *R_c_* is acquired by one of the four channels of the ADC; DAC2 provides the sensor film voltage ($V_{i}$); *R_f_* sets the IOA gain whose output (SIGNAL) is sampled by a second ADC channel.

**Figure 3 nanomaterials-13-02549-f003:**
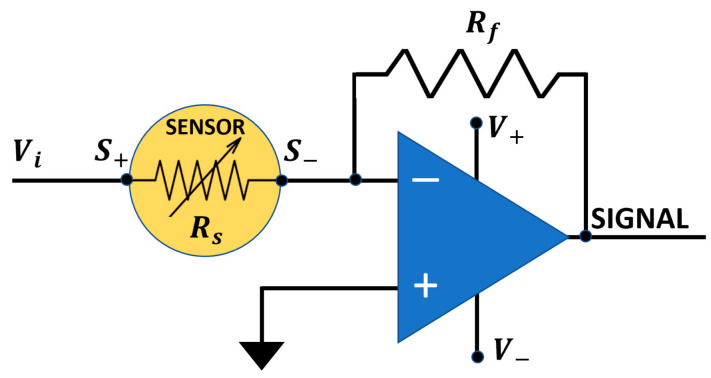
Sketch of the sensor signal transduction circuit. $V_{i}$ is the sensor film feeding voltage, $S_{+}$ and $S_{-}$ are the sensor film pins, $R_{f}$ is the IOA feedback resistor, SIGNAL is the sensor voltage output, and V_+_ and V_−_ are the IOA power supply.

**Figure 4 nanomaterials-13-02549-f004:**
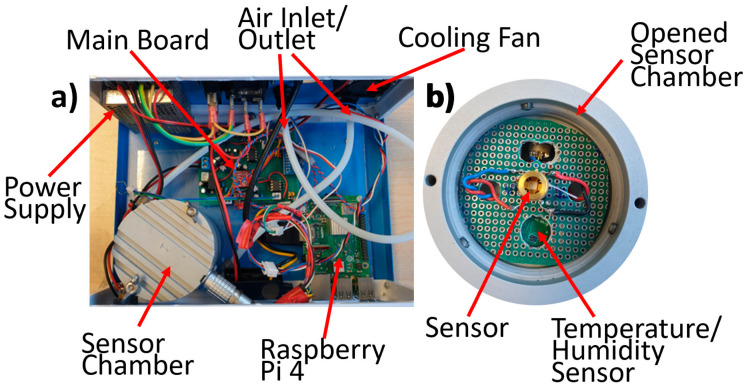
Photograph of the device interior. (**a**) The entire device interior hosting the sensor chamber, the sensor board, the Raspberry Pi 4, and the 24 W power supply; (**b**) enlargement of the top view of the opened chamber showing the board hosting the tested sensor; the humidity/temperature is visible through a hole drilled in this board.

**Figure 5 nanomaterials-13-02549-f005:**
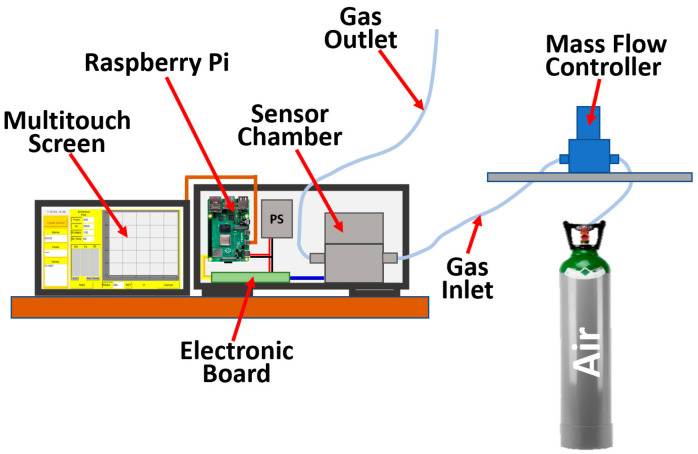
Sketch of the device laboratory set-up. PS is the device power supply; the mass flow controller conveys the air from (for instance) a cylinder to the sensor chamber; the electronic board manages the sensor heating and the data acquisition; the custom software acquires, stores, and plots the data in real time on the external multitouch screen.

**Figure 6 nanomaterials-13-02549-f006:**
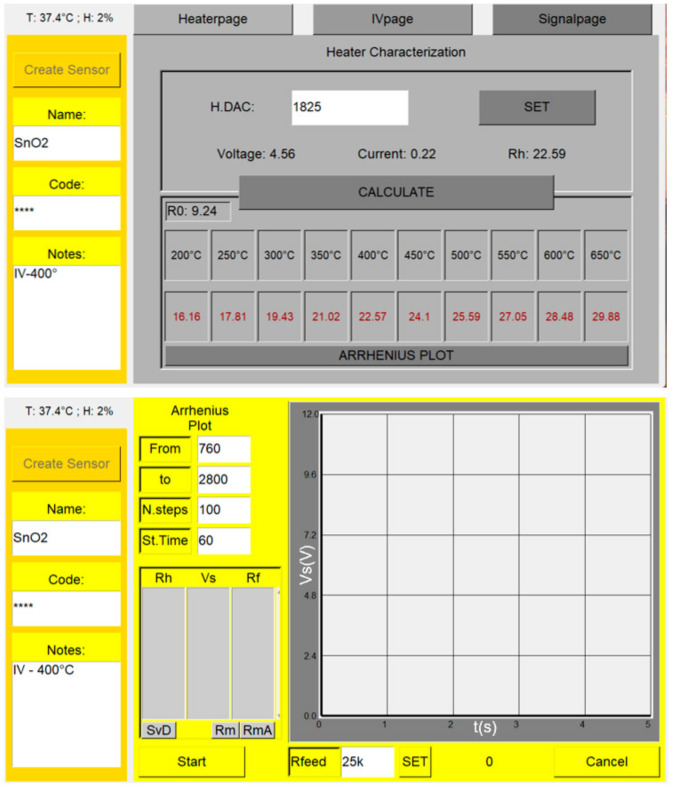
Screenshot of the pages handling the sensor heating and the Arrhenius plot.

**Figure 7 nanomaterials-13-02549-f007:**
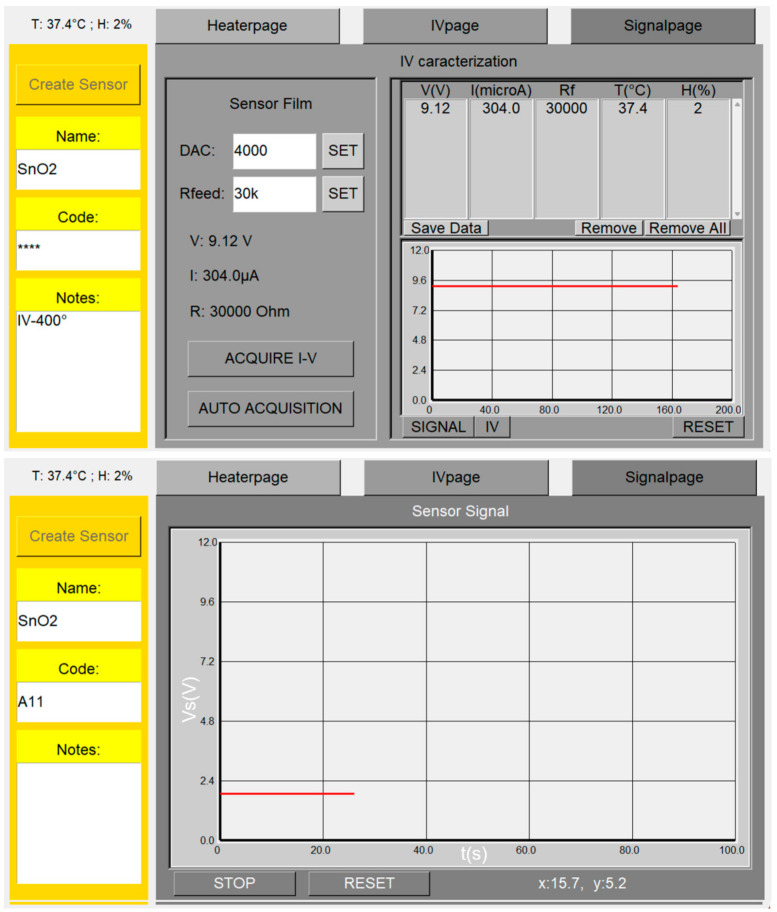
Screenshot of the pages handling the I–V and the sensor output.

**Figure 8 nanomaterials-13-02549-f008:**
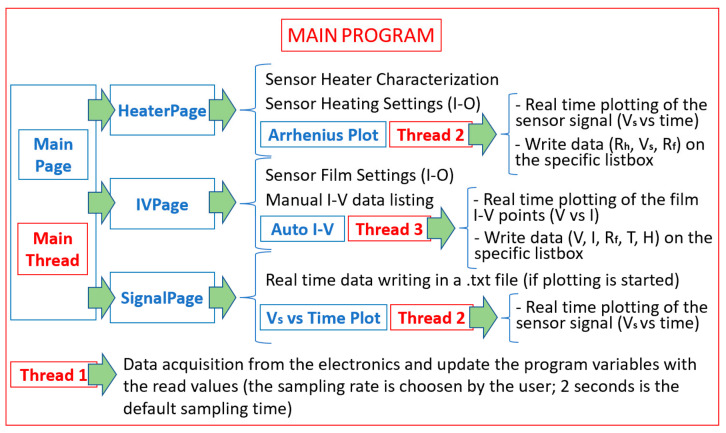
Block scheme of the main program handling the I–V plot, the Arrhenius plot, and the sensor output. Blue and red squared blocks represent the Graphical User Interface (written by exploiting the Tkinter module of Python) pages and the software threads, respectively.

**Figure 9 nanomaterials-13-02549-f009:**
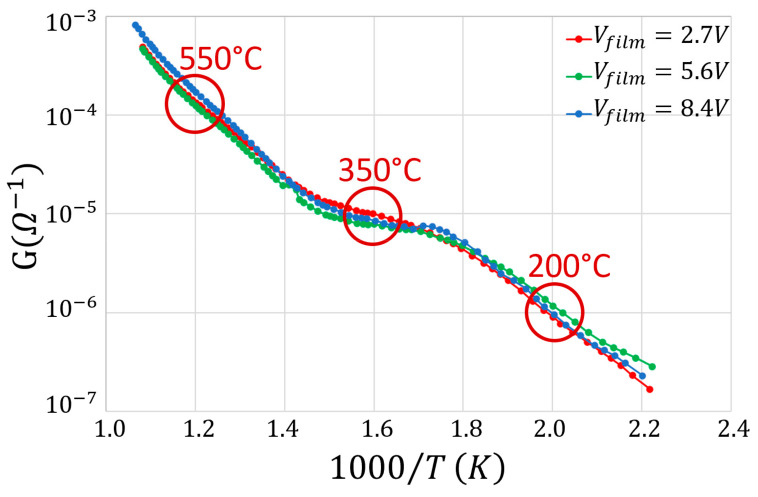
Arrhenius plot of SnO_2_ + 1%Au sensor in dry air. Upper panel: plot of the conductance (G, in a logarithmic scale) versus the reciprocal of the absolute temperature (T) at three different film voltages; the red circles indicate the temperatures (200, 350, and 550 °C) at which the I–Vs were taken (see below); lower panel: the average Arrhenius plot for the three voltages (black line) and the +/− standard errors (gray lines).

**Figure 10 nanomaterials-13-02549-f010:**
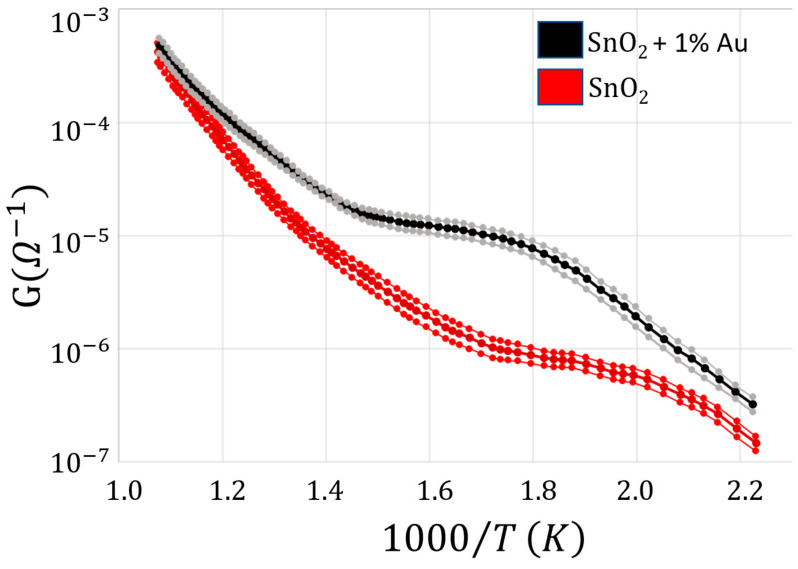
Comparison between the average Arrhenius plots of SnO_2_ and SnO_2_ + 1%Au in dry air. The Arrhenius plot for the three voltages in Figure 9 (upper panel) is repeated three times. Each are averaged together (nine total readouts for each point) for SnO_2_ (dark red thick line) and SnO_2_ + 1%Au (black thick line); for clarity, the +/− standard errors are represented by the two red and gray lines above and below the average ones, respectively.

**Figure 11 nanomaterials-13-02549-f011:**
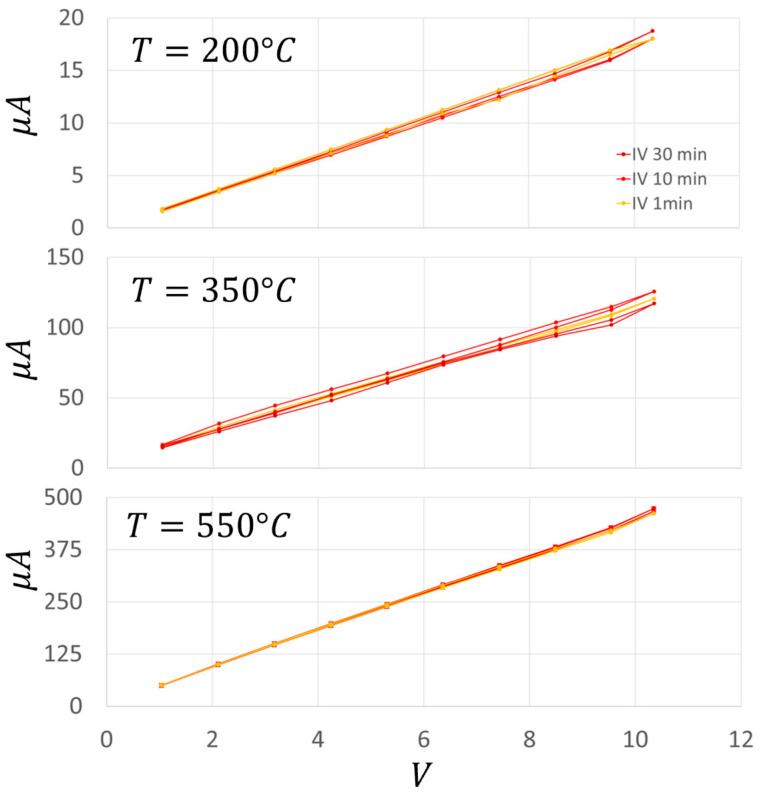
Current to voltage (I–V) characteristics of SnO_2_ + 1%Au sensor. I–V characteristics recorded for voltages ranging from 0 to 10.4 V in 1.04 V steps at 200 °C (**top panel**), 350 °C (**middle panel**), and 550 °C (**lower panel**) in ascending and descending directions; the steps lasted 1 min (orange), 10 min (red), and 30 min (dark red).

**Figure 12 nanomaterials-13-02549-f012:**
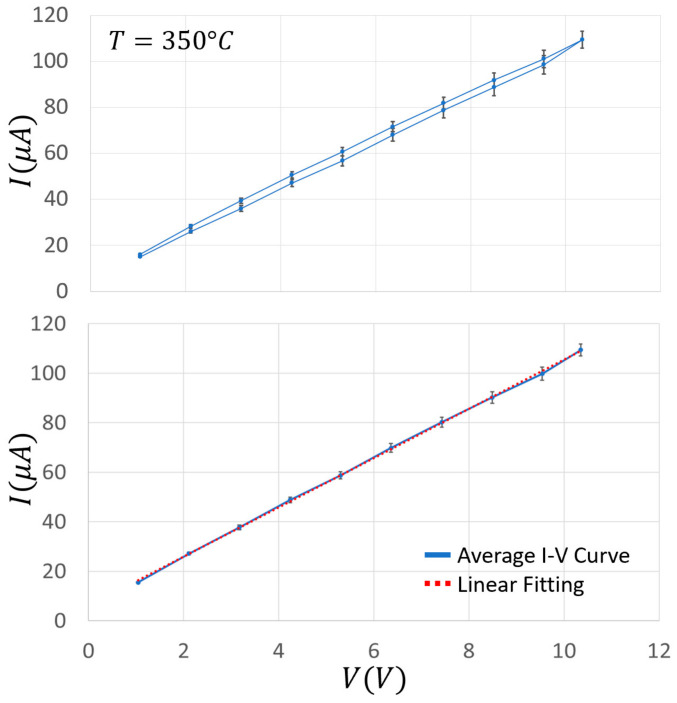
Average I–Vs of SnO_2_ + 1%Au sensor at 350 °C. I–V curves recorded for voltages ranging from 0 to 10.4 V in 1.04 V steps at the WT of 350 °C; each data point of the top panel is the average of three current measurements to voltage steps in ascending and descending directions that lasted 1, 10, and 30 min, each one repeated three times; the bottom panel groups all the data points (*n* = 171) related to the same film voltage (no matter if it was taken at ascending or descending voltages); the data were linearly fitted by the equation: I=a·V+b, where a = 9.95 μA/V and b = 6.02 μA.

**Figure 13 nanomaterials-13-02549-f013:**
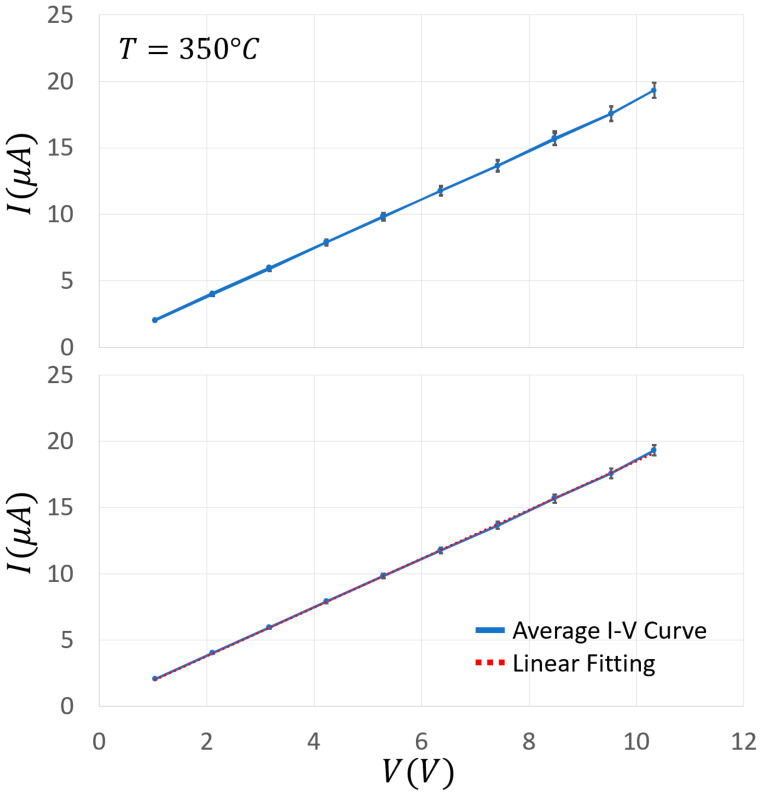
Average I–Vs of SnO_2_ sensor at 350 °C. I–V curves were recorded for the same voltage range, step, and durations of Figure 12; the bottom panel groups all the data points (*n* = 171) as in Figure 12; the data are linearly fitted by the equation: I=a·V+b, where a = 1.84 μA/V and b = 0.01 μA.

## Data Availability

Data are available under request.

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
