# Peer review of "A Portable Device for I–V and Arrhenius Plots to Characterize Chemoresistive Gas Sensors: Test on SnO2-Based Sensors"

_nanomaterials, 2023, doi:10.3390/nano13182549_

Round 1
Reviewer 1 Report
The paper reports on approach for MOX sensor calibration.
The approach makes novel use of low cost technology, including a Raspberry PI computer and DIY hardware.
There is useful background on MOX sensors and the rational for the work.
The experimental setup is clearly described.
Characterisation results are presented which demonstrate how the system can be used to characterise a gas sensor in dry air, although not with other gases.
There is maybe too much detail on the software. Perhaps this section could be better summarised.
Line 23: This sentence needs clarifying.
Line 32: MOX sensors are not known for their stability, so suggest removing this comment.
Line 38: Larger MOX particles can also be used for sensing.
Line 54: I think 'overwhelming' is the wrong word.
How where the analogue signal digitised? Was a separate ADC used?
Section 2.5: What module is used to develop the GUI?
Can the authors suggest how the setup could be modified to investigate the response to different gases and humidity? I believe more MFCs and other hardware would be needed.
I assume the setup does not have environmental temperature control?
Minor editing is required to improve the English.
Reviewer 2 Report
The presented article is aimed at the development of modern compact nanostructured gas sensors. The device consists of a small sealed chamber (containing a sensor socket and a temperature/humidity sensor), a pneumatic system, and special electronics.
SnO2 and SnO2+1%Au sensors based on the commonly used tin oxide material were selected for device testing and validation.
The article is of particular interest. However, there are the following questions.
1. It is necessary to expand the introductory part and indicate the current work aimed at the development of new methods for analyzing the composition of gas mixtures and the development of compact analyzers.
See for example works:
Zhou Ch., em all. Determination of organic impurities by plasma electron spectroscopy in nonlocal plasma at intermediate and high pressures //Plasma Sources Science and Technology. - 2022. - Vol. 31. - no. 10. – P. 107001
Zhou C., em all . Using Collisional Electron Spectroscopy to Detect Gas Impurities in an Open Environment: CH4-Containing Mixtures // Molecules. - 2022. - Vol. 27. - no. 18. - S. 6066.
2. Questions related to selectivity and sensitivity limits of the developed sensors remained open, that is, what are the minimum proportions of impurities that can be detected?
3. Apparently, the sensors allow you to determine only a qualitative analysis of impurities in the gas. This is true? Are there any prospects in determining the quantitative analysis of the composition of gas mixtures?
After eliminating these shortcomings, the article can be accepted for publication in the journal.
